# A Novel Bandstop Filter Based on Two-Port Coaxial Cavities for the Installation of Metallic Mode Stirrers in Microwave Ovens

Juan Monzo-Cabrera *, Alejandro Díaz-Morcillo [ID], Antonio Martínez-Gonzalez [ID], Antonio Lozano-Guerrero [ID], José Fayos-Fernández [ID], Rafael Pérez-Campos [ID] and Juan Luis Pedreño-Molina [ID]

Departamento de Tecnologías de la Información y las Comunicaciones, ETSI de Telecomunicación, Universidad Politécnica de Cartagena, Plaza del Hospital, 1, E-30302 Cartagena, Spain; alejandro.diaz@upct.es (A.D.-M.); toni.martinez@upct.es (A.M.-G.); antonio.lozano@upct.es (A.L.-G.); jose.fayos@upct.es (J.F.-F.); rafael.perez@upct.es (R.P.-C.); juan.pmolina@upct.es (J.L.P.-M.)
* Correspondence: juan.monzo@upct.es; Tel.: +34-968326510

**Abstract:** Metallic stirrers are commonly used in mode-stirred reverberation chambers but are less common in microwave ovens, because the use of a metallic axis creates a coaxial port in the cavity wall where the axis is introduced, which may result in significant microwave leakage levels. However, due to electromagnetic, mechanical, temperature, or chemical requirements, metallic axes must be used in some cases. In this paper, we present a high power coaxial filter that enables the use of metallic stirrers in microwave applications. The filter employs two-port coaxial cavities that are analyzed to determine their design parameters. These cavities are coupled via coaxial lines to achieve proper attenuation levels. The cavities were designed using commercial electromagnetic (EM) software, and a filter prototype was validated using vector network analyzer (VNA) measurements. The simulated and measured results show that a three-stage configuration can achieve attenuations greater than 70 dB at the 2.45 GHz ISM band, allowing the use and external handling of the metallic axis inserted into microwave ovens with negligible microwave radiation leakage.

**Keywords:** microwave heating; metallic mode stirrer; microwave applicator; coaxial filter; coaxial cavity





## 1. Introduction

Multimode cavities with one or more magnetrons are commonly used in industrial microwave-heating applications. Because of multiple mode combinations, these microwave cavities generate electric field standing-wave patterns [1], which may result in cold and hot spots in the processed materials. Uneven electric field distribution can result in nonuniform temperature patterns and thermal runaway inside materials due to a variety of factors including microwave feeding location, material dielectric and thermal properties, cavity and material geometry, material location inside the oven, and sample movement [2].

There are several techniques that can reduce hot and cold spots within multimode microwave cavities and provide more uniform heating such as the use of dielectric multi-layer structures [3], variable or selected frequencies [4,5], or a combination of phase delay between two or more sources [6], mainly when solid-state generators are employed [7]. However, the most commonly used techniques for increasing uniformity in the electric field pattern are based on product movement, using a turntable in domestic ovens [8,9] or a conveyor belt in industrial tunnel applicators, and time-variant modification of the applicator geometry with mode stirrers [10–12].

In the case of turntables or mode stirrers, the movement is generated by an external motor and transmitted through a rotary axis that penetrates the cavity. This also happens in reverberation chambers, where mode stirrers are often employed to emulate different wireless propagating environments but only with low power signals. This movement

transmission necessitates the applicator being open in the zone where the axis passes through the cavity walls, which can result in microwave radiation leakage.

Figure 1 presents the cross-section scheme of a multimode microwave applicator where a dielectric material is heated. Metallic stirrers are typically preferred over dielectric stirrers due to their greater ability to reflect and disperse electromagnetic fields; therefore, a metallic axis capable of supporting the stirrer blades may be required. Unfortunately, because there is no cutoff frequency for the transversal electromagnetic modes (TEM), this metallic axis creates a coaxial port in the cavity wall where it is introduced; thus, bandstop filters are required to avoid leakage of microwave radiation.

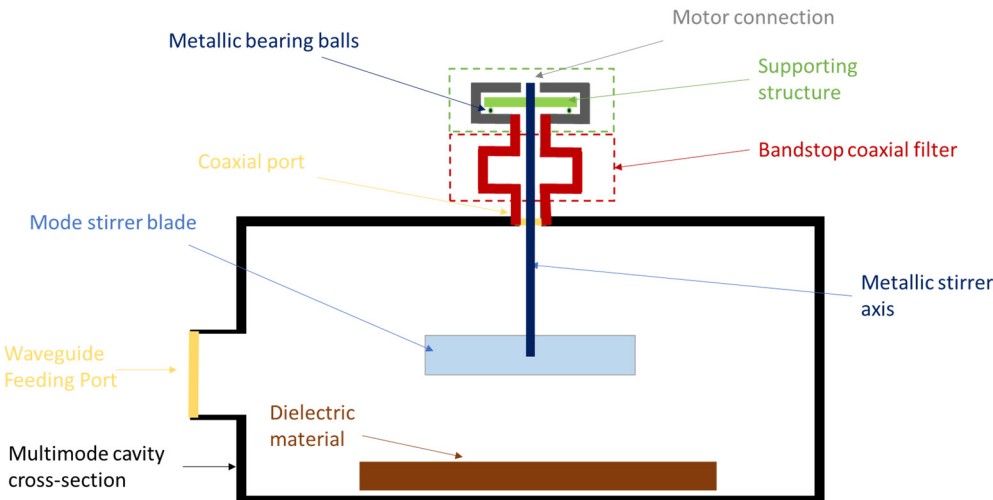

**Figure 1.** Cross-section scheme for the installation of metallic mode stirrers in multimode microwave-heating applicators with filtering and rotating structures.

Metallic bearing balls can be used to allow the stirrer to rotate while also preventing microwave leakage. These metallic balls are frequently placed beneath a metallic supporting structure, and this metallic contact provides additional electric field shielding if properly designed. However, when the metallic bearing balls make contact with the cavity walls without any intermediate bandstop filter, high currents may appear as a result of the structure's movement, causing the balls and the supporting structure to overheat and eventually degrade. To reduce electric field leakage and current intensities at mobile structures, a coaxial filter should be used when the mode stirrer axis is metallic.

Microwave bandstop filters are frequently used in microwave heating ovens. This is the case with corrugated filters [13–15] at the input and output ports in tunnel applicators, or $\frac{\lambda_g}{4}$ chokes at the oven door (where $\lambda_g$ is the wavelength at the operation frequency) [16]. High-pass filters with cutoff waveguide grids can also be found in microwave oven doors or at air flow inlets or outputs. In all of these cases, an appropriate design must be developed in order to satisfy the EN-55011 standard [17] for the electromagnetic compatibility of industrial, scientific, and medical (ISM) equipment in Europe.

Microwave heating applications use high power narrow-band signals operating at one of the reserved ISM bands, most commonly 915 MHz or 2.45 GHz, but also 433.92 MHz, 5.8 GHz, or 24.125 GHz. This leads to the creation of bandstop filters centered on one of these frequencies.

To the best of the authors' knowledge, no work on coaxial filters that allow the use of metallic-axis mode stirrers in microwave applications has been reported. This paper presents a novel design for a 2.45 GHz bandstop filter based on two-port coaxial cavities to avoid microwave leakage and ensure the proper operation of moving parts inside the applicator, such as metallic mode stirrers. First, the behavior of individual resonators is analyzed in relation to their design parameters; then, the final geometry and performance of a three-stage filter are simulated using EM software and validated with measurements

using a vector network analyzer. An eccentricity analysis is also performed to predict the behavior of the proposed filter when the metallic stirrer axis moves nonideally.

This paper builds on our previous publication [18]. The following are the major new contributions in this paper:

- A two-port coaxial cavity-based novel filter structure is presented, analyzed, and validated.
- An eccentricity analysis is performed, demonstrating that the proposed filter can function properly even when the filter axis, which corresponds to the mode stirrer axis, displaces or tilts, versus its ideal central rotating position.

## 2. Use of Coaxial Cavities to Implement Microwave Bandstop Filters

Bandstop filters have been created by concatenating resonators with microwave lines, as previously reported in [19,20]. In [19], for instance, the authors demonstrated how to connect resonators with waveguide or microstrip lines to create narrow stopband filters. A cascade of aperture-coupled coaxial cavity bandstop filters was used in [20] to achieve up to 55 dB attenuation levels over a wide frequency range. In [21], coaxial cavities were combined with CPW lines to form a two-pole bandstop filter with a continuously tunable center frequency.

In this work, we use two-port high power coaxial cavities as resonators and coaxial lines to couple them in a cascade configuration. Sun et al. [22] demonstrated that an equivalent lumped model of coaxial cavities can be realized using a parallel tank of resistance, inductance, and capacitance, and that this type of cavity can function as a bandstop filter. In [23], a two-port resonant coaxial stub was analyzed, and designing curves for band-pass filters was shown to aid in the design process.

Figure 2 illustrates a cross-section of a coaxial cavity with two coaxial lines serving as feeding and transmitting ports. The ports allow for the mode stirrer to be rotated as well as coupled to other coaxial cavities. Thus, the cavity's coupling structures are two identical coaxial rings with $a_1$ and $b_1$ outer and inner radios, respectively. Furthermore, we designed $a_1$ and $b_1$ to provide 50 Ω impedance in the feeding and inner connecting coaxial lines, ensuring impedance continuity with SMA measuring cables and used transitions. The CST Studio Suite was used to optimize the cavity's height, $L_2$, and radius, $b_2$, to provide the best attenuation at the resonant frequency.

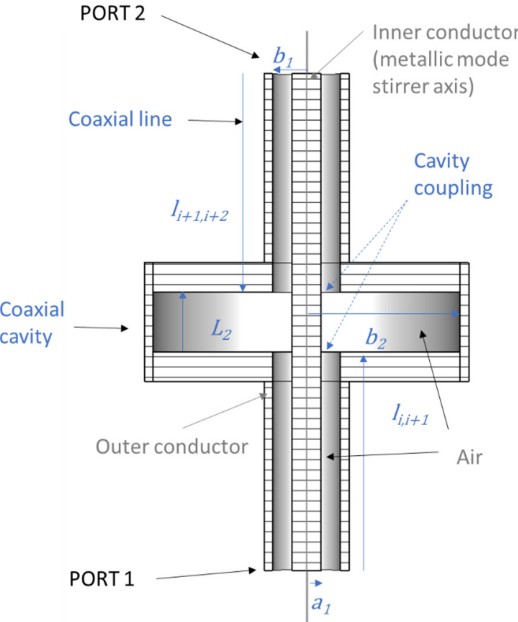

**Figure 2.** Two-port coaxial cavity employed as a bandstop filter resonator.

To provide adequate attenuation levels across the 2.4–2.5 GHz ISM band, the central frequencies of the coaxial cavity resonators were designed to be 2.45 GHz, and the cavities were coupled with short coaxial lines, as illustrated in Figure 3.

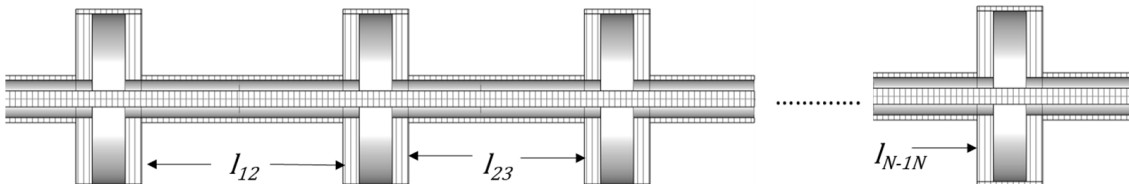

**Figure 3.** Cross section of the coupling of $N$ coaxial resonators, $l_{ij}$ being the intracavity coupling lines.

## 3. Materials and Methods

### 3.1. Procedures for Design and Simulation

The design procedure was solely based on the use of an EM simulator. To determine whether the resonator element proposed in Figure 2 had bandstop features or not, a simple trial and error procedure was used at first. This hypothesis was investigated because step-impedance filters can be used for this purpose, as previously demonstrated [24]. After determining that a two-port coaxial cavity could be used to reject a frequency band, we performed several parametric sweeps for the inner radio ($b_2$) and height ($L_2$) to identify a range of values capable of generating a resonant frequency at 2.45 GHz. Finally, within that range, we used an optimization process to estimate the $b_2$ and $L_2$ values that produced the maximum attenuation at that central frequency. The final design of the filter was achieved by coupling several cavities, as shown in Figure 3, with $l_{ij}$ being the intracavity thickness that allowed for coupling adjustment.

CST Studio Suite 2020 [25] was employed to simulate the coaxial structures used as bandstop resonators or filters. The frequency solver, with adapting mesh capabilities, was applied in the frequency band ranging from 2 to 3 GHz. The calculations were carried out on a Legion Desktop-VI648NN with 16 Gbytes of RAM Memory, an Intel Core i7-9750HF CPU, and the Windows 10 operating system.

### 3.2. Simulation Scenarios

Figure 4 shows an axial section of a structure made up of a 3D coaxial cavity model fed by two coaxial lines. This structure was simulated in the CST Studio Suite, and two coaxial ports were used to excite the structure's TEM mode. All simulations were carried out using perfect electric conductors (PEC) and vacuum materials. With this bandstop resonator configuration, the metallic mode stirrer axis could freely rotate, as shown in Figure 4. This structure was subjected to parametric sweeps and optimization procedures.

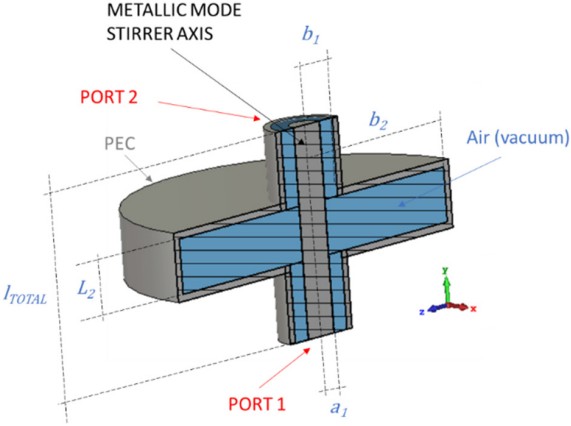

**Figure 4.** Simulated structure for the coaxial cavity bandstop resonator. $l_{TOTAL}$ = 6 cm, $b_1$ = 1 cm, and $a_1$ = 0.5 cm. The $b_2$ and $L_2$ values were fixed using parametric sweeps or using optimized values.

Figure 5 depicts the coupling of three optimized coaxial cavities separated by 4 mm intracavity lines, resulting in a compact design with higher attenuation levels than a single resonator design. Finally, Figure 6 illustrates a cross section of the three-stage filter with DIN 7/16 transitions. This scenario was created in order to observe the effect of the transitions required to perform the measurements.

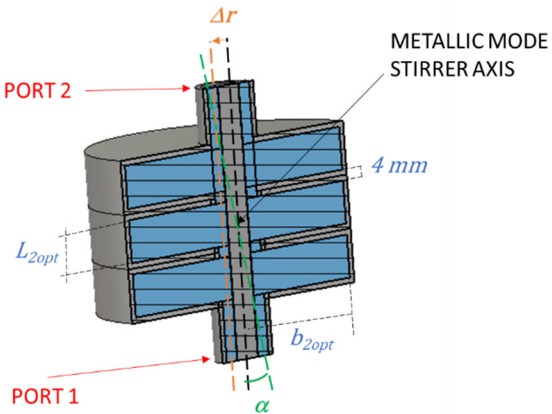

**Figure 5.** Cross section showing a compact coupling of three optimized coaxial cavities with an internal 4 mm coaxial line separation. $l_{ij}$ = 4 mm. $\Delta r$ and $\alpha$ were employed for the eccentricity analysis.

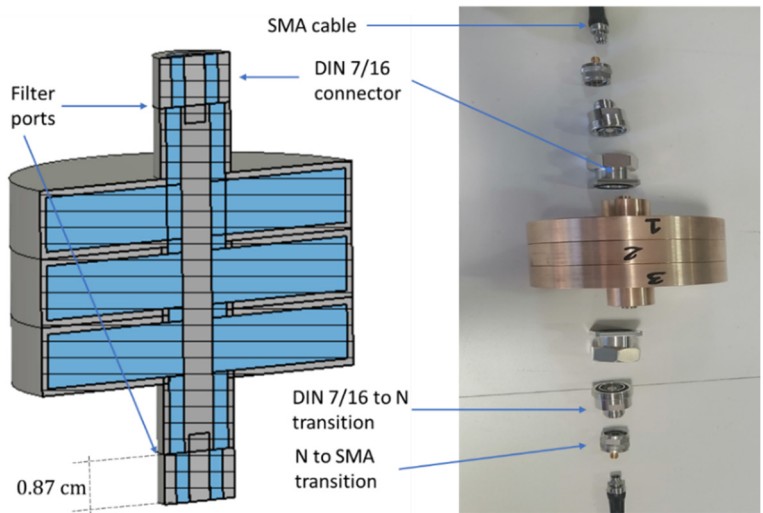

**Figure 6.** Transition and connectors needed for the connection of the VNA ports to the filter ports.

### 3.3. Procedures for Optimization

The interpolated Quasi-Newton optimization strategy [26] was used in the CST Studio Suite optimizer to calculate both the optimal coaxial-cavity height ($L_{2opt}$) and inner cavity radius ($b_{2opt}$) of the coaxial cavity resonator, depicted in Figure 4. The goal was to achieve the lowest possible value for the $S_{21}$ parameter at 2.45 GHz. The $L_2$ and $b_2$ optimal search intervals were [4.8, 5.3] and [1.4, 1.6] (cm), respectively.

### 3.4. Eccentricity Analysis Parameters

Since mode stirrers are typically moving within the filter, as depicted in Figure 5, several nonideal situations may arise, such as displacement of the mode stirrer axis from the filter central position or inclination of the stirrer axis from the ideal position, denoted as $\Delta r$ or $\alpha$ in Figure 5. Both eccentricity situations were simulated in this work to better understand their impact on the proposed filter. These situations were simulated using parametric sweeps for $\Delta r$ and $\alpha$. Specifically, $\Delta r$ was swept from 0 to 0.4 cm in 0.1 steps and $\alpha$ from 0 to 4° in 1° steps.

### 3.5. Measurement Configuration

The S21 scattering parameter of the three-stage coaxial filter, shown in Figure 6, was measured using a Rohde & Schwarz ZVA67 Vector Network Analyzer (VNA) in the 2–3 GHz frequency range to validate the proposed design of the bandstop filter.

Because both the internal coaxial conductor, which corresponds to the mode stirrer metallic axis, and the external coaxial conductor of the designed filter ports did not match the usual dimensions of coaxial cables and connectors, several coaxial connectors and transitions were used to connect the filter ports to the VNA ports.

Figure 6 depicts the scheme for connecting the filter port to the DIN 7/16 male connector, as well as the manufactured filter and all of the connectors and transitions used. As shown in Figure 6, the DIN 7/16 connector allowed one to thread its inner coaxial conductor to the stirrer axis and connect the external conductor of the DIN 7/16 connector to the external conductor of the filter's coaxial port. The N and SMA transitions allowed us to keep 50 $\Omega$ while avoiding unwanted reflections. The VNA was calibrated in this manner at the SMA cable connectors using conventional standards and procedures. The intermediate filter (IF) bandwidth was set to 1 kHz, and 10,001 points were used within the frequency range.

## 4. Results

### 4.1. Simulation Results

Initially, parametric sweeps in the CST Studio Suite were used to analyze the main design parameters of a single two-port coaxial resonator, specifically the inner radius ($b_2$) and height ($L_2$). Figure 7a represents the variation of the resonant frequency when the cavity height was held constant at $L_2$ = 1.559 cm, and $b_2$ varied from 4.8 to 5.25 cm. Based on the simulations, the higher the $b_2$ values for a given stirrer axis radius, the lower the resonant frequency. In fact, with a 0.45 cm variation in $b_2$, the resonant frequency could vary by more than 250 MHz around the 2.45 GHz ISM central frequency. Figure 7b shows the relationship between resonant frequency and coaxial cavity height ($L_2$). In this case, $b_2$ was fixed at 5.126 cm, while $L_2$ varied [1.4, 1.6] (cm). As expected, larger heights resulted in lower resonant frequencies in the cavity. It should be noted that, as shown in Figure 7a,b, this single resonator provided a minimum attenuation of 20 dB for a bandwidth of 100 MHz, and that at least three stages should be used if a filter with a 60 dB attenuation level was required. Furthermore, based on the results in Figure 7, it is possible to conclude that both $L_2$ and $b_2$ can be optimized to provide maximum attenuation in the 2.4–2.5 GHz ISM band.

The simulated S21 scattering parameter of an optimized coaxial cavity is shown in Figure 8. The optimal cavity dimensions in this case were $L_{2opt}$ = 1.6 cm, and $b_{2opt}$ = 5.132 cm, which were obtained using the EM software's Quasi-Newton optimizer after 800 steps, including interpolations. The optimized structure had a maximum attenuation level of about 70 dB at 2.45 GHz and lower attenuation levels of about 20 dB at the ISM band's extremes.

### 4.2. Analysis of Eccentricity

Figure 9 demonstrates the three-stage filter's behavior when the mode stirrer axis remained in an ideal position, $\Delta r = 0$, or when it deviated from the filter center, $\Delta r$ cm, due to unwanted vibrations or swaying, as shown in Figure 5. According to the results in Figure 9, the ideal three-stage filter could attenuate more than 120 dB in the center of the band and more than 70 dB across the entire 2.45 GHz band. The displacement of the stirrer axis, on the other hand, caused an increase in the resonant frequency and a widening of the filter bandwidth. Fortunately, even in extreme cases where the axis almost touched the coaxial port's outer conductor, the filter provided attenuation levels greater than 60 dB.

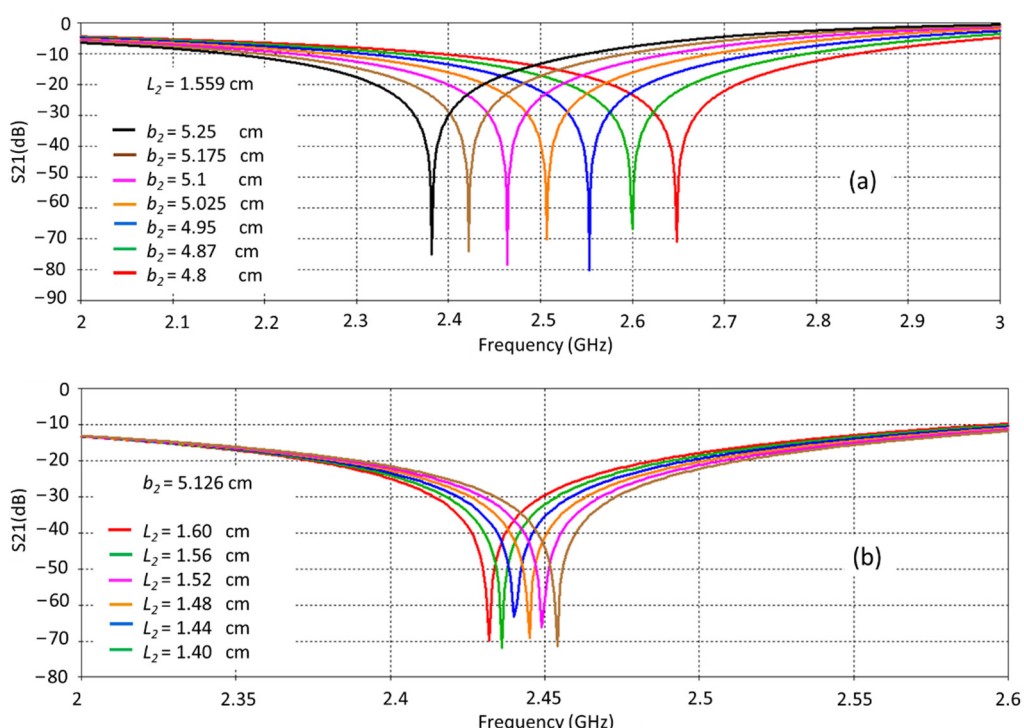

**Figure 7.** (**a**) Resonant frequency variation versus cavity inner radius. (**b**) Dependence on cavity height.

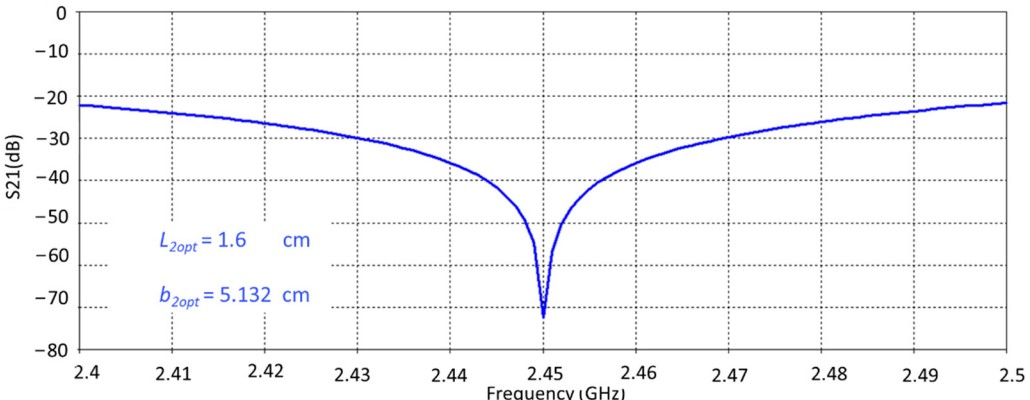

**Figure 8.** Simulated results of the optimized coaxial cavity acting as a bandstop filter. $L_{2opt}$ = 1.6 cm, $b_{2opt}$ = 5.132 cm.

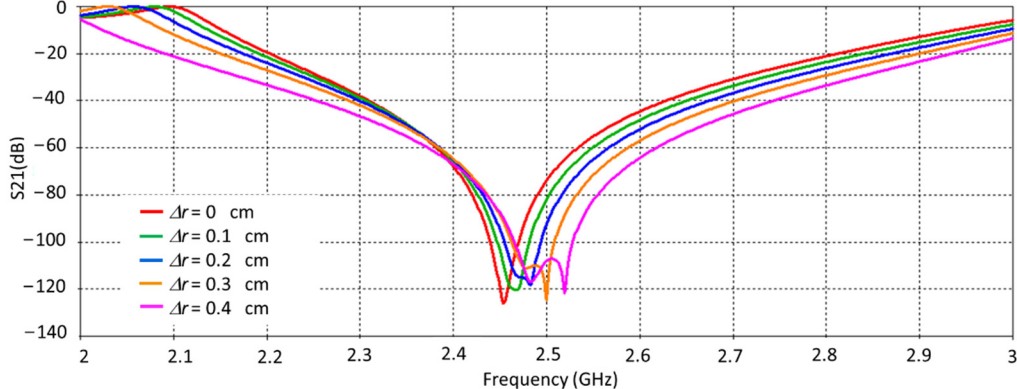

**Figure 9.** Effect of the mode stirrer axis's displacement ($\Delta$r) from its ideal position.

Figure 10 depicts the effect of the mode stirrer axis inclination ($\alpha$) versus the ideal situation shown in Figure 5. The simulations showed that the greater the inclination angle, the larger the shift of the resonant frequency. Although small changes in the filter bandwidth were perceived, they were less noticeable than those caused by variations in $\Delta r$. In this case, too, the attenuation levels were kept above 60 dB for the 2.4–2.5 GHz band.

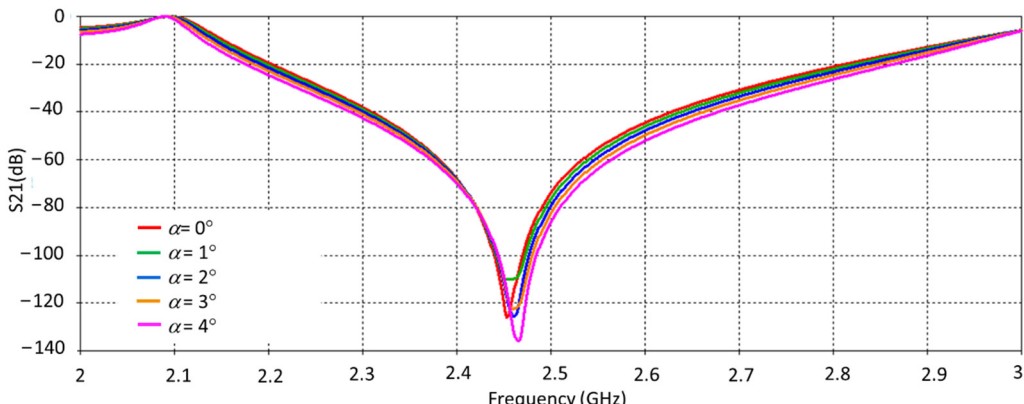

**Figure 10.** Effect of the mode stirrer axis's inclination ($\alpha$) from its ideal position.

Consequently, the eccentricity effects should be negligible for this type of filter, at least in the 2.45 GHz band, indicating that these filters can be used safely even under the extreme machinery vibration or swaying conditions found in industrial applications.

### 4.3. Validation of the Proposed Filtering Structure

Figure 11 compares the simulation results of the bandstop filter obtained by coupling three optimized coaxial cavities, all with the dimensions $L_{2opt}$ = 1.6 cm and $b_{2opt}$ = 5.132 cm, as shown in Figures 5 and 6, to the measurements performed with the VNA.

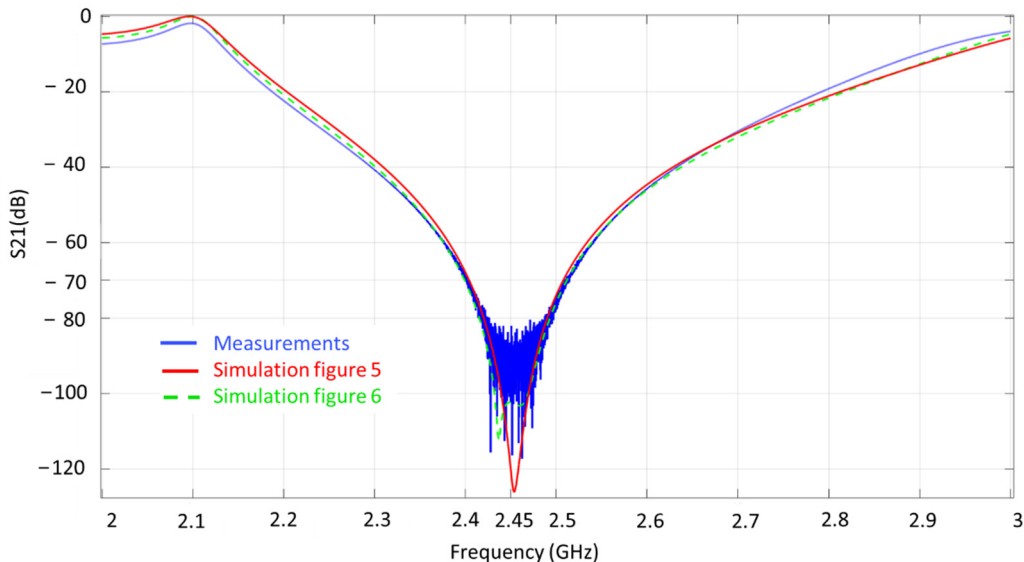

**Figure 11.** Three-stage coaxial-cavity filter validation: measurements (blue), simulations without transitions (red), and simulations with DIN 7/16 connectors (green).

It should be noted that the results showed that the DIN 7/16 connectors had no discernible effect on filter behavior, because both the filter and the connectors had an intrinsic impedance of 50 $\Omega$. It can also be noticed that the simulations and measurements agreed very well, and that the attenuation levels greater than 70 dB were obtained in the 2.4–2.5 GHz ISM band. As a consequence, this filter would provide sufficient attenuation

levels to comply with the EN55011 standard while also protecting rotating bearing balls from overheating currents. The filter would also allow for the introduction and rotation of the metallic mode stirrer axis.

### 5. Discussion

This contribution analyzed and validated a new bandstop filter based on the coupling of two-port coaxial cavities that allows the introduction of metallic cylinders inside microwave cavities. This novel design opens up new possibilities for the design of microwave cavities, particularly in applications requiring metallic stirring of the material and/or electromagnetic fields.

The analysis of the single coaxial resonators revealed that the resonant frequency could be easily adjusted and optimized by varying the coaxial cavity's height and/or radius. However, the results showed that single cavities could only provide up to 20 dB of attenuation along the 2.45 GHz ISM band, requiring several stages to be coupled to achieve higher bandstop rejection levels.

As a result, a three-stage filter was designed, manufactured, and validated. Both simulations and measurements agreed very well without the need for any tuning elements, with attenuation levels greater than 70 dB for the entire [2.4, 2–5] GHz band and around 100 dB at 2.45 GHz. When considering the addition of metallic bearing balls and metallic anchorage structures to achieve a more robust stirrer rotation, rejection levels may be increased.

The eccentricity analysis suggested that the filter behavior would not deviate significantly from the ideal situation and would provide more than 60 dB attenuation levels in the stop band even in extreme cases of axis displacement or inclination caused by machine vibrations and movements. Thus, this type of filter appears to be very well suited to the use of metallic mode stirrers in any working environment.

Although the filter is very compact, it could be made even smaller by using low-loss ceramic or plastic dielectrics within the cavities and coaxial lines. It should be noted that while this filter was designed specifically for 2.45 GHz microwave heating systems, similar results can be extrapolated to other ISM frequencies. In fact, in recent years, a combination of frequencies has been used to improve the material's heating uniformity. As a result, multi-band-stop filters could be designed in situations involving multiple frequency sources. Further research in this area is planned.

**Author Contributions:** Conceptualization, J.M.-C. and A.D.-M.; methodology, J.M.-C. and A.D.-M.; software, J.M.-C. and A.D.-M.; validation, A.M.-G., A.L.-G., J.F.-F. and R.P.-C.; formal analysis, J.L.P.-M.; investigation, J.F.-F.; resources, J.M.-C.; data curation, A.M.-G.; writing—original draft preparation, J.M.-C. and A.D.-M.; writing—review and editing, J.M.-C., A.D.-M., J.L.P.-M., A.M.-G., A.L.-G., J.F.-F. and R.P.-C.; visualization, J.M.-C.; supervision, J.M.-C. All authors have read and agreed to the published version of the manuscript.

**Funding:** This research was partially funded by Fundación Séneca, Agencia Regional de Ciencia y Tecnología de la Región de Murcia, with the 21640/PDC/21 grant, entitled, "Sistema de Radiación y filtrado para tambores rotatorios en aplicaciones directas para secado de textiles por microondas (ROTDRY)".

**Institutional Review Board Statement:** Not applicable.

**Informed Consent Statement:** Not applicable.

**Data Availability Statement:** Not applicable.

**Conflicts of Interest:** The authors declare no conflict of interest.

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
