# Peer review of "A Novel Bandstop Filter Based on Two-Port Coaxial Cavities for the Installation of Metallic Mode Stirrers in Microwave Ovens"

_electronics, doi:10.3390/electronics11131989_

Round 1

Reviewer 1 Report

The Article is devoted to the design and study of high power 2.45 GHz coaxial band-stop filters that makes it possible a use of metallic-axis mode stirrers in microwave devices. The filters prevent leakage of microwaves and ensure the correct operation of the moving part. The novelty of work is related to the fact that such filters have not been used up to now.

The behavior of individual resonators has been analyzed depending on the design parameters such as the inner radius and height. The characteristics of optimized three-stage filter were calculated in the final geometry using EM simulators. Experimental measurements were carried out using a vector network analyzer, which confirmed the calculated dependences. An eccentricity analysis is also carried out to predict the behavior of the proposed filter with non-ideal movement of the axis of the metal stirrer. Simulated and measured results show that using a three-stage configuration of coaxial filter can give very high attenuations of microwave radiation leakage higher than 70 dB at the 2.45 GHz ISM band.

The investigations presented in Article show that even for extreme cases of axis displacement or inclination due to machinery vibrations and movements the filter will provide more than 60 dB attenuation levels in the stop band.

The Article present new results which are actual and of interest for Electronics journal. The Article can be published as it is.

Reviewer 2 Report

This article proposes a solution to prevent leakage due to the use of metal stirrers in high power microwave applicators. This solution, quite innovative, consists in forming coaxial cavities playing the role of a stop band filter at the working frequency of the applicators. The authors show good results obtained by electromagnetic simulations and by measurements carried out on a prototype. Sensitivity analyzes were also presented.

However, the design method of the stop band filter is not explained, and the reviewer finds it difficult to understand the link between the diagram presented in Fig.3 and the configuration in Fig.5. From my point of view, each cavity must present a series resonance behaving as a short circuit at the working frequency of the applicator, and this "short circuit" will be in shunt connection with the main line which is the coaxial line; the intra-cavity thicknesses allow adjustment of the coupling between the 3 cavities. The reviewer considers Fig.3 no conform to the physical reality of the structure.

As the objective of the filter, the authors invoke the attenuation necessary to satisfying the EN55011 standard. Some data from a realistic scenario will help better understand the problem.

Reviewer 3 Report

The authors present a high-power coaxial filter that allows the usage of metallic stirrers in microwave applicators. The filter with two-port coaxial cavities  are analyzed to understand the design parameters and connected through coaxial lines to obtain proper attenuation levels. Generally, the paper is well written and  can be accept in the cuurent version. The only point is that the language should be polished further.
